# Real-Life Vancomycin Therapeutic Drug Monitoring in Coagulase-Negative Staphylococcal Bacteremia in Neonatal and Pediatric Intensive Care Unit: Are We Underestimating Augmented Renal Clearance?

**DOI:** 10.3390/antibiotics12111566

**Published:** 2023-10-26

**Authors:** Claudia Sette, Marcello Mariani, Luca Grasselli, Alessio Mesini, Carolina Saffioti, Chiara Russo, Roberto Bandettini, Andrea Moscatelli, Luca A. Ramenghi, Elio Castagnola

**Affiliations:** 1Department of Pediatrics, Ospedale SS. Annunziata, 74121 Taranto, Italy; 2Pediatrics and Infectious Diseases Unit, IRCCS Istituto Giannina Gaslini, 16147 Genoa, Italy; 3Pediatric Emergency Room and Emergency Medicine, IRCCS Istituto Giannina Gaslini, 16147 Genoa, Italy; 4Division of Infectious Diseases, Department of Health Sciences (DISSAL), University of Genova, 16132 Genoa, Italy; 5Central Laboratory of Analysis, IRCCS Istituto Giannina Gaslini, 16147 Genoa, Italy; 6Neonatal and Pediatric Intensive Care Unit, IRCCS Istituto Giannina Gaslini, 16147 Genoa, Italy; 7Neonatal Intensive Care Unit, IRCCS Istituto Giannina Gaslini, 16147 Genoa, Italy

**Keywords:** vancomycin, PK/PD, TDM, augmented renal clearance

## Abstract

Bloodstream infections (BSI) from coagulase-negative-staphylococci (CoNS) are among the most frequent healthcare-related infections. Their treatment involves the use of vancomycin, a molecule whose optimal pharmacokinetic/pharmacodynamic (PK/PD) target for efficacy and safety is an area-under-curve/minimum inhibitory concentration (AUC/MIC) ratio ≥ 400 with AUC < 600. BSIs from CoNS in pediatric and neonatal intensive care unit that occurred at the Gaslini Institute over five years were evaluated to investigate the efficacy of vancomycin therapy in terms of achieving the desired PK/PD target and determining whether any variables interfere with the achievement of this target. AUC/MIC ≥ 400 with AUC < 600 at 48 and 72 h after therapy initiation was achieved in only 21% of the neonatal population and 25% of the pediatric population. In the pediatric population, an inverse correlation emerged between estimated glomerular filtration rate (eGFR) and achieved AUC levels. Median eGFR at 72 h was significantly higher (expression of hyperfiltration) in events with AUC < 400, compared with those with AUC ≥ 400 (*p* < 0.001). A cut-off value of eGFR in the first 72 h has been identified (145 mL/min/1.73 m^2^), beyond which it is extremely unlikely to achieve an AUC ≥ 400, and therefore a higher dose or a different antibiotic should be chosen.

## 1. Introduction

Healthcare-related infections (HAIs) present a major public health problem resulting in prolonged hospitalization, rising hospital care costs and increased mortality since the spread of multi-resistant microorganisms [1]. The expression “ICU-acquired-HAI” refers to healthcare-related infections in intensive care unit (ICU), occurring in patients with invasive devices (e.g., endotracheal tube, vascular and urinary catheters) that were placed or manipulated 48 h before infection onset [2,3]. Bacteremia or bloodstream infection (BSI) are defined by the presence of at least one blood culture (single or repeated) positive for a microorganism recognized as pathogenic with the patient presenting infection symptoms such as fever (>38 °C), chills, or hypotension. BSI could also be defined in the presence of “skin contaminants” in patients with symptoms and two positive blood cultures for Coagulase-negative staphylococci (CoNS) or other skin contaminants (e.g., *Micrococcus* spp., *Propionibacterium acnes*, *Bacillus* spp., *Corynebacterium* spp.—https://www.cdc.gov/nhsn/xls/master-organism-com-commensals-lists.xlsx, accessed on 19 July 2023) that developed from two different blood samples collected within 24 h. In newborns, especially preterm and/or low birth weight, a single blood culture could be sufficient [4]. According to the most recent ECDC report, published in 2019, 8.3% of patients admitted to Intensive Care Units in Europe had at least one care-related infection in 2017 [5]. In 45% of cases it was a BSI (3.7%). CoNS, *Staphylococcus aureus* and *Escherichia coli* represent the three pathogens most commonly associated with pediatric HAIs [6]. The management of ICU-acquired BSIs involves the timely initiation of empirically reasoned intravenous antibiotic therapy. In general, it is indicated to initiate therapy that provide anti-Gram-positive and anti-Gram-negative spectra of action. For the former group, antistaphylococcal penicillins or cephalosporins are of choice, with the addition of vancomycin, teicoplanin or daptomycin [7] in cases of high prevalence of methicillin-resistant *Staphylococcus aureus* (MRSA) and CoNS. Vancomycin is an antibiotic belonging to the glycopeptide class, with bactericidal activity, that acts by inhibiting cell wall peptidoglycan synthesis. Its spectrum of action is limited to Gram-positive cocci and bacilli, including beta-lactamase-producing and methicillin-resistant staphylococci (both MRSA and CoNS) Vancomycin has high molecular weight, is water-soluble and very stable in acid solution. The protein binding is variable but does not exceed 50–55%, has a half-life of about 4–6 h (in adults), and is eliminated renally. The potential toxic effects are renal and otovestibular [8]. Based on its pharmacokinetic and pharmacodynamic (PK/PD) characteristics, vancomycin is a time-dependent antibiotic with prolonged post-antibiotic effect, so the most useful PK/PD parameter for assessing its efficacy is the area under the time–concentration curve/minimal inhibitory concentration (AUC/MIC) ratio [8], that according to guidelines [9,10], should be ≥400 and <600, ideally within 24–48 h after initiation of therapy, at least in severe MRSA infections. Unfortunately, these parameters are not well defined/identified for methicillin-resistant (Met-R) CoNS. The calculation of AUC during intermittent administration is performed via an equation [11] that requires knowledge of the baseline level (C_min_) and the level at the end of administration (C_max_), whereas in the case of continuous infusion, AUC can be calculated by multiplying the steady-state concentration by 24 (a factor that corresponds to the interval in hours between doses, which is 24 in the case of continuous infusion). In this case, the steady-state concentration corresponds to the blood level of drug measured at any time, provided that at least 12 hours have elapsed since the start of continuous infusion therapy. To achieve the AUC_24h_ target blood level of vancomycin must range between 17 and 25 mg/L. Following these indications, in the case of MIC for vancomycin > 1 the possibility of achieving PK/PD targets without exceeding toxic levels is impossible. Consequently, the use of alternative molecules should be considered. The purpose of our study is to describe the pharmacokinetics of vancomycin in a cohort of ICU pediatric patients with Met-R CoNS BSI, the attainment of PK/PD targets, and determine whether any variables could interfere with the achievement of these parameters.

## 2. Results

From January 2016 to December 2020, 153 episodes of Met-R CoNS bacteremia occurred in 140 patients. Appendix A shows etiology distribution events. Appendix A shows vancomycin MIC distribution in CoNS isolates over study years. In 58% (89/153) MIC for vancomycin was ≤1 mg/L (72 MIC = 1 mg/L and 17 MIC = 0.5 mg/L) while in the remaining 42% of isolates (64/153) MIC = 2 mg/L was detected. No strain was resistant to vancomycin. Most of the events (90/153; 59%) occurred in the neonatal population, with minor variations in terms of frequency over years, which were not significant (Appendix A).

### 2.1. Neonatal Population

In the neonatal population, 90 events occurred in 82 patients, with 32 (39%) being females. The median age at diagnosis was 15 days (25–75th centile: 9–21). Clinical and microbiological characteristics are summarized in Appendix A. In 9 out of 90 events (10%), the patient died within 30 days. In all those cases, death occurred in premature and cardiopathic patients and no correlation with BSI was found. No reinfection at 15 days was documented. In 71/90 (79%) events, at least one vancomycin plasmatic level was determined within 48 h after therapy commencement, in 80/90 (89%) within 72 h, and in the remaining 10 events, the first determination was obtained 72 h after therapy commencement (Figure 1). Laboratory and pharmacological parameters are summarized in Table 1. AUC/MIC ≥ 400 with AUC < 600 was reached in 15/71 events (21%). A significant linear correlation between age (in days) and eGFR was found (Spearman rho 0.661, *p* < 0.001). A significant correlation was also found between mean albumin at 48 h and AUC at 48 h (Pearson r 0.288, *p* = 0.028) but no correlations were found between AUC and eGFR. The population was then grouped between median AUC < 400 and ≥400 within the first 48 and 72 h of treatment: mean albumin levels were higher in those with AUC > 400 at 48 h. There were no other statistically significant differences between the two groups. Data are summarized in Table 2.

### 2.2. Pediatric Population

In the pediatric population, 63 events occurred in 58 patients, with 25 (43%) being female. Median age at diagnosis was 21 months (25–75th percentile: 2.5–133). Clinical and microbiological characteristics are summarized in Appendix A. In 1 out of 63 events (1.6%), patient died within 30 days. In all those cases, death was unrelated to BSI. Reinfection was recorded in 1 case (1.6%). In 57/63 (90%) events, at least one blood level of vancomycin was determined within 48 h after therapy commencement, in 61/63 (97%) within 72 h, and in the remaining 2 events, the first determination of vancomycin plasmatic level was obtained after 72 h since therapy commencement (Figure 1). Laboratory and pharmacological parameters are summarized in Table 1. In the first 48 h of therapy, the number of events with AUC/MIC ≥ 400 and AUC < 600 was 14/57 (25%). The same result was observed extending the analysis to the first 72 h of vancomycin treatment: 15/61 (25%) events had AUC/MIC ≥ 400 with AUC < 600. A statistically significant inverse correlation emerged between median AUC and median eGFR both in first 48 h (Pearson’s r: −0.295, *p* = 0.049) and 72 h (Pearson’s r: −0.353; *p* = 0.012) of vancomycin therapy (Appendix A). By grouping population between AUC < 400 and ≥400 in the first 48 h of vancomycin therapy, median eGFR levels were found to be statistically different, confirming the inverse correlation between these two variables. In the first group (AUC < 400), the median eGFR was 150 mL/min/1.73 m^2^ while in the latter (AUC ≥ 400) eGFR had a lower value of 95.2 mL/min/1.73 m^2^ (*p* = 0.025). Significance was maintained also extending the analysis to the first 72 h of treatment. Data are expressed in Table 2. An ROC curve was then calculated to identify a cut-off value of eGFR beyond which AUC ≥ 400 is unlikely to be achieved. A median eGFR value in the first 72 h of 145 mL/min/1.73 m^2^ was found to be predictive of failure in achieving AUC ≥ 400 (sensitivity 59.46%, specificity 92.31%, Youden index (informedness) 0.518, positive likelihood ratio of 7.73 and negative likelihood ratio of 0.44), Appendix A.

## 3. Materials and Methods

### 3.1. Study Design

We conducted a retrospective, single-center study at the IRCCS Istituto Giannina Gaslini in Genoa, Italy. Events of Met-R CoNS bacteremia in ICU patients treated with vancomycin that occurred from 1 January 2016 to 31 December 2020 were analyzed.

### 3.2. Standard of Care

Following the indications provided by AIFA (Agenzia Italiana del Farmaco—Italian drug agency) in their technical data sheet, at our institute vancomycin is prescribed at a dosage of 10 mg/kg in a one-hour infusion as a loading dose, followed by 40 mg/kg/day as a continuous infusion for invasive methicillin-resistant staphylococcal infections. At least 12 h after the initiation of therapy, dosing of vancomycin plasma levels is performed to adjust the posology aiming to achieve AUC/MIC ≥ 400 with AUC < 600.

### 3.3. Inclusion Criteria

Inclusion criteria were: hospitalization at the Pediatric Intensive Care Unit (PICU) or Neonatal Intensive Care Unit (NICU) for more than 48 h at the time of infectious episode/blood culture specimen collection; isolation of methicillin-resistant CoNS on at least 2 blood cultures taken from two separate blood samples and spaced no more than 24 h apart in patients > 30 days old; isolation of methicillin-resistant CoNS on single blood culture in patients ≤ 30 days old; vancomycin susceptibility of isolated strain, with MIC recording; antibiotic therapy with vancomycin and determination of at least one plasmatic level during treatment. In the case of a second event in the same patient: new positive blood culture for CoNS at least 15 days after the end of therapy for the previous event. Pathogen identification and antibiotic susceptibility testing were performed using automated systems MALDI-TOF (bioMérieux, Marcy l’Etoile, France) and Sensititre (Thermo Scientific, Milan, Italy). MICs were interpreted according to EUCAST criteria [12]. Determination of vancomycin blood levels was obtained using a commercial biochemical immunoassay with an automated analyzer (Roche C501; Roche, Milan, Italy).

### 3.4. Data Collection

Anonymized demographic data were collected: age, sex, biometric parameters (weight, height), baseline disease, presence of central line if any, 30-day outcome in terms of mortality, 15-day outcome in terms of reinfection (positivity for the same germ on blood culture during treatment, after one or more negative blood cultures), and all determinations of creatinine and albumin levels during antibiotic therapy with vancomycin. As creatinine was measured by enzymatic IDMS (isotope dilution mass spectroscopy) validated method, estimated glomerular filtration rate (eGFR) was calculated by modified Schwartz bedside formula (update 2009) for all patients regardless of age [13]. If multiple creatinine dosages were collected within the same day, only the worst was considered. Microbiological data were collected (date of blood sample collection, microorganism, vancomycin MIC) as well as details regarding vancomycin therapy: start date, plasmatic levels, AUC and AUC/MIC in the first 48 h and 72 h after initiation of therapy. Vancomycin levels “within” 48 h were collected in any case not less than 24 h after therapy commencement, so that a steady state is unlikely to have been achieved. At our center, antibiogram is usually available no sooner than 72 h after blood culture specimen collection. We therefore decided to consider only AUC value (and not the AUC/MIC ratio) since MIC data are usually not yet available in the crucial stages of the first 48–72 h of therapy. The chosen AUC threshold was 400: this value represents the necessary cut-off value in the treatment of CoNS BSIs amenable to vancomycin therapy (i.e., with MIC ≤ 1) to achieve the desired PK/PD target of AUC/MIC ≥ 400. Considering the extreme heterogeneity of patient ages, neonatal and pediatric populations were evaluated separately. The former included all patients up to 30th day of life and the latter all patients from 31 days of life to 18 years.

### 3.5. Statistical Methods

Descriptive statistical analysis was performed for population, clinical and microbiological data. For continuous variables with a normal distribution (identified by the Shapiro–Wilk test), mean and 95% confidence intervals (95% CI) were presented, while for those with a nonnormal distribution, median, 25th and 75th percentiles were calculated. Absolute values and percentages were used for categorical variables. To compare different groups, parametric or nonparametric tests were used depending on variable distribution. A *p*-value less than 0.05 was considered statistically significant. To identify a threshold level of eGFR above which it is unlikely to reach an AUC value ≥ 400, an analysis using “receiver operating characteristics” (ROC) curve and calculation of likelihood ratio and Youden (informedness) index was performed. All tests were performed using statistical software Jamovi 2.3.28, a graphical R frontend (https://www.jamovi.org/, accessed on 29 August 2023).

## 4. Discussion

In our study, we analyzed the epidemiology of Met-R CoNS BSI and PK/PD parameters of vancomycin for their treatment that occurred in children hospitalized in PICU and NICU at the Giannina Gaslini Institute over a five-year period. We decided to evaluate CoNS BSI because these are the most frequent infections in NICU and PICU at our center [14] and therefore have the greatest impact in terms of vancomycin therapy. On the epidemiological side, most of the events involved the neonatal population, and the most frequently isolated CoNS was *Staphyloccoccus epidermidis*. Although in more than half of the events vancomycin MIC of CoNS was ≤1 mg/L, an important percentage (42%) had MIC = 2 mg/L, the breakpoint indicated by EUCAST. Albeit considered susceptible according to EUCAST criteria, staphylococci with this MIC for vancomycin probably should be treated with alternative antibiotics, as suggested for MRSA, given the impossibility of achieving the desired efficacy target (AUC/MIC ≥ 400) without leading to toxicity (AUC < 600). These data regarding local epidemiology show that, at our institute, use of vancomycin as empirical therapy in critical patients with blood culture development of Gram-positive cocci could not be adequate, since more than 40% of CoNS BSI might need alternative therapies. We also analyzed PK/PD of vancomycin in our patient populations. In the critically ill patient, vancomycin administered as continuous infusion is considered best for achieving effective concentrations [10]. In our series, following AIFA indications on vancomycin starting dose, the therapeutic target of efficacy and toxicity (AUC/MIC ≥ 400 and <600) at 48 and 72 h after therapy commencement was achieved in only 21% of neonates and 25% of other ages. The dosages in the vancomycin technical datasheet provided by the Italian drug agency (10 mg/kg every 6 h or 10 mg/kg in 1-h loading dose and 40 mg/kg/day in continuous infusion) seem utterly insufficient to achieve pharmacodynamic targets, suggesting the use of higher dosages such as those provided by the 2020 ASHP-IDSA guidelines (60–80 mg/kg/day in patients under 3 months and 60–70 mg/kg/day in patients > 12 years) [10]. This unsatisfactory result is in contrast with the favorable outcome found in our study with only one reinfection and no infection-related mortality. This apparent incongruity may be explained by the fact that the PK/PD target was inferred from severe MRSA infections guidelines and could not be appropriate for Met-R CoNS. Due to CoNS reduced pathogenicity, a lower AUC/MIC ratio could be sufficient [15,16]. Two main variables (serum albumin and eGFR) were then analyzed to assess whether they might affect achievement of the PK/PD target. In the neonatal population, subjects who reached an AUC ≥ 400 at 48 h after therapy commencement had significantly higher mean albumin values than those who did not reach the AUC target. this finding correlates with changes in albumin binding of vancomycin in the neonatal period. Several studies have in fact demonstrated how vancomycin albumin binding differs among adult, pediatric, and neonatal populations, with the latter showing higher levels of unbound fraction [17,18]. As with other hydrophilic molecules, vancomycin is mainly distributed in the extracellular space, and the unbound fraction is renally excreted.

The higher quota of unbound drug results therefore in an augmented volume of distribution (Vd) and, consequently, augmented renal excretion, reducing AUC accordingly. PK is not the only parameter that is altered due to variable albumin binding: the drug unbound fraction is the pharmacologically active one, thus performing an AUC from plasmatic levels of total drug (bound and unbound) may not be adequate to estimate PD of vancomycin. Additional challenges in estimating vancomycin PK/PD in the neonatal population are thus represented by complex interactions between multiple factors: increased drug unbound fraction (and consequently, pharmacologically active molecule), and augmented renal excretion but reduced GFR due to maturational changes in the nephrons. All these elements probably result in different PK/PD targets for vancomycin therapy in neonates (probably lower than the “usual” AUC/MIC > 400). This difference could possibly explain the low rate of reinfection and infection-related deaths after vancomycin therapy in neonates despite the lack of attainment of “general population” AUC/MIC targets. In this population, therefore, the lack of PK/PD target attainment, more than augmented renal clearance, could be due to numerous other factors: the reduced albumin binding mentioned above, the administration of underdosed vancomycin, or the premature timing of monitoring samples, before achieving the steady state. The complex interaction between these factors could be better assessed by dosing the free drug quota (rather than the total) and delaying, if possible, the execution of drug sampling.

In the pediatric population, on the other hand, an inverse correlation emerged between estimated renal function and achieved AUC levels, both in the first 48 and 72 h of treatment. In patients with median AUC < 400 both within 48 h and 72 h of therapy, median eGFR was statistically higher (as an expression of hyperfiltration) than those with median AUC ≥ 400. This effect can be explained by the participation of mainly two factors: the relatively low vancomycin protein binding and the increased renal clearance in critically ill patients: pediatric patients could respond to an acute inflammatory illness with an increase in renal clearance leading to augmented drug elimination and consequential undertreatment [19]. In our study, this could translate to increased urinary elimination of unbound vancomycin, resulting in lower plasma levels and consequently decreased AUC. These results are consistent with those of other studies in both pediatric and critically ill adult cohorts in which the association between augmented renal clearance and failure to reach the AUC target was documented [20,21].

According to our findings, in a critical pediatric patient older than 30 days, on empirical vancomycin therapy for Gram-positive cocci bacteremia, with a GFR estimated by Schwartz formula greater than 145 mL/min/1.73 m^2^ in the first 72 h of therapy, vancomycin therapy has an extremely high probability (positive predictive value greater than 95%) of failing to achieve the desired PK/PD target. Under these conditions, it might then be necessary to either administer higher vancomycin doses or switch to another molecule, less affected by renal function. If augmented renal clearance phenomenon has an impact in vancomycin PK/PD in low severity events such as CoNS BSI, this may be even more important in *S. aureus* BSI. Furthermore, given the yearly increase in strains with MIC > 0.5 for vancomycin already observed in our center [22], Staphylococcal BSI empiric therapy should be reconsidered using different vancomycin dosages or different molecules whose PK/PD is less affected by renal clearance and maturational changes.

## Figures and Tables

**Figure 1 antibiotics-12-01566-f001:**
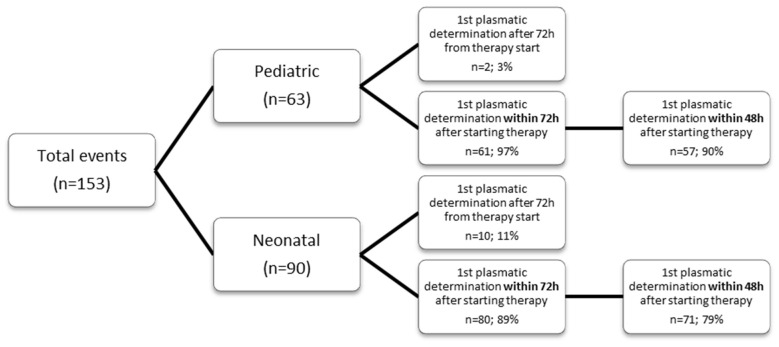
Event distribution according to vancomycin plasmatic levels timing.

**Table 1 antibiotics-12-01566-t001:** Laboratory and pharmacological parameters; ^1^ according to Schwartz modified formula.

	48 h	72 h	48 h	72 h
	Neonatal Population		Pediatric Population	
**Median AUC (25–75°)**	389 (280–486)	369 (307–484)	328 (270–404)	333 (274–415)
**Median AUC/MIC (25–75°)**	269 (172–412)	267 (176–416)	276 (180–382)	268 (168–353)
**Mean albumin: mg/dL (95th CI)**	3323 (3173–3473)	3362 (3210–3514)	3363 (3212–3514)	3403 (3265–3542)
**eGFR ^1^ median: mL/min/1.73 m^2^ (25–75°)**	38 (27–57)	37 (24–57)	126 (82–205)	131 (87–212)
**AUC ≥ 400 (%)**	34 (48)	36 (45)	15 (26)	18 (30)
**AUC/MIC ≥ 400 (%)**	20 (28)	22 (28)	14 (25)	15 (25)
**AUC ≥ 600 (%)**	6 (8)	6 (8)	1 (2)	0 (0)
**AUC/MIC ≥ 400 with AUC < 600 (%)**	15 (21)	17 (21)	14 (25)	15 (25)

**Table 2 antibiotics-12-01566-t002:** Median AUC associated variables in neonatal and pediatric population; ^1^ according to Schwartz modified formula.

	Neonatal Population						Pediatric Population					
	48 h			72 h			48 h			72 h		
	*AUC < 400*	*AUC ≥ 400*	*p value*	*AUC < 400*	*AUC ≥ 400*	*p value*	*AUC < 400*	*AUC ≥ 400*	*p value*	*AUC < 400*	*AUC ≥ 400*	*p value*
**Mean albumin (mg/dL)**	**3147**	**3487**	**0.037**	3226	3484	0.089	3348	3400	0.752	3381	3451	0.643
median eGFR (mL/min/1.73 m^2^) ^1^	39.1	34.1	0.986	39.2	32.7	0.839	**150**	**92.5**	**0.025**	**159**	**85.6**	**0.001**

## Data Availability

Data are available on request from the corresponding author.

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
