# Peer review of "Real-Life Vancomycin Therapeutic Drug Monitoring in Coagulase-Negative Staphylococcal Bacteremia in Neonatal and Pediatric Intensive Care Unit: Are We Underestimating Augmented Renal Clearance?"

_antibiotics, 2023, doi:10.3390/antibiotics12111566_

Round 1

Reviewer 1 Report

Comments and Suggestions for Authors

-Line 21: Change "bloodstream infections" into "Bloodstream infections".

-The affiliation should be added for the postcode, and the order number of the institute should be in uppercase.

-The materials and methods part should be divided into subtopics or subsections for easier understanding and accessibility.

-Please revise the consistency and correct the font and the format (such as "." and the space between the words, etc.) throughout the manuscript.

-The symbols (such as "//") in the table should be described under the table.

Comments on the Quality of English Language

 Moderate editing of English language required.

Author Response

Dear reviewer

Thanks for your precious comments. Hereby provided point-by-point responses.

1) corrected

2) edited as suggested

3) materials and methods was divided as suggested

4) symbols were removed

Manuscript was also revised for format and english language.

Best regards

Marcello Mariani

Reviewer 2 Report

Comments and Suggestions for Authors

the author try to evaluate in their paper in a Real-life  setting the vancomycin therapeutic drug monitoring in coagulase- 3
negative staphylococcal bacteremia in neonatal and pediatric in- 4
tensive care unit the one. THis is of the most intriguing issue in infetious disease

Paper is well designed I would suggest only  to add some more references and sicuss about these in the paper

Minor criticism

Please add paper like:"The relationship between vancomycin AUC/MIC and trough concentration, age, dose, renal function in Chinese critically ill pediatric patient from Chen et al" or else  "Vancomycin therapeutic monitoring by measured trough concentration versus Bayesian-derived area under the curve in critically ill patients with cancer."

Improve discussion based on the above mentioned references

Comments on the Quality of English Language

minor spell check

Author Response

We thank the reviewer for the interesting papers suggested.

We added references and consequently enhanced the discussion.

Best regards

Marcello Mariani

Reviewer 3 Report

Comments and Suggestions for Authors

I have read this paper with great interest, and value the topic and the effort made. However, I do have major concerns on the current (pooled) analysis. I have provided my comments chronologically, but the major issues relate to dose use, the absence of reflection on maturational changes in clearance, maturational protein binding aspects, and eGFFR thresholds. 

First, the dose assessed (10 mg/kg loading dose, and 40 mg/kg/day continuous infusion) is quite low, especially if targeted in both nicu and picu cases, so that the overall low target AUC attainment (indeed extrapolated form staph aureus pneumonia in adults, to CONS sepsis) is likely much more explained by the low doses applied. When verifying eg the FDA label, a higher initial dose is suggested (15 mg/kg), followed by 20-40 mg/kg/day in preterms, or at least 40 mg/kg/day in children. If indeed a single dose has been used, it is very reasonable to expect that the AUC observed will mainly be driven by maturational differences> augmented clearance. Related to this, the conclusion in the full paper is more reasonable versus the abstract (as besides other drugs, we could also alter the dose).

This lower loading dose may result in delayed ‘steady state’, so we need more information (figure 1) on the determination of vanco concentrations ‘within 48 h’ as the text suggests sampling from 12 h onwards ?

I miss a reference to teicoplanin in the introduction.

The PK characteristics described in the intro are likely from adults ? and not yet supported by a reference ? (suggestion is to consider the Colin et al paper, as describing the PK throughout human life, PMID 30656565).

How have data been handled if e.g. several creatinine values are available, was the highest or lowest selection, or all observations ? how has creatinine been measured, and was the assay IDMS validated ?

The eGFR for the Schwartz and the Flanders formula a very different, but I do not understand how and why the authors use the Flanders formula, as references 12 and 13 are not related to the Flanders formula (Pottel et al, Pediatr Nephrol ?), but has never been claimed nor studied in neonates or children (age range 1.6-8.7 year). Reference 13 has assessed different function, specific for vancomycine clearance, and suggests that the Schwartz is more useful ? Please reconsider this part of the paper, and better explain the approach taken. I would suggest to consider to remove the Flanders approach, unless I have misunderstood the methods as applied (what do you mean with Flanders meta data ?)

Related to glomerular hyperfiltration, there is a very useful paper on threshold definitions (Pottel et al, Pediatr Nephrol 2023, PMID 36459244).

Why combine both NICU and PICU cases, and if so, why use the 30 days PNA as cut off, as 44 postmenstrual age (PMA) is much more relevant if preterms are included ? (cf check Ward et al, Pediatr Res 2017, PMID 28248319)

Table 1 and Table 2: please check (mg/dl) the albumin denominator ? 3147 mg/dl ? perhaps g/dL is more commonly used ?

We do need information on the maturational differences (weight, age) in median AUC values (as it is very reasonable to assume that the AUC will mainly depend on these covariates, and might be impacting the Pearson analyses reported.

Discussion section

I do not understand the reasoning described in lines 226-228.

Free fractions and or concentrations of vancomycin in neonates and children have been described, and this information should be incorporated in the paper, and in the analysis and interpretation of the data (cf Smits et al, Eur J Clin Microbiol Infecti Dis 2018, PMID 29770901; Oyaert et al, Antimicrob Agents Chemother 2015, PMID 26349820, or Leroux et al, Br J Clin Pharmacol 2019, PMID 30834552). I’m aware that this might be perceived as ‘product placement’, but these aspects are simply relevant for this paper.

Author Response

We gratefully thank the reviewer for the precious comments that, in our opinion, contributed to further enhance our paper.

  • We totally agree with the reviewer about the (too) low vancomycin dose administered both as loading and as maintenance. However, as this is a retrospective study, it is based on the indications that were in force at the time of data collection. Prior to this study, in fact, the dosage prescribed at our Institute was that indicated by AIFA (Italian Drug Agency) in vancomycin technical data sheet (i.e. 40mg/kg/day for pediatric population). Thanks to reviewer comment we therefore better explained in standard of care section and in discussion. Abstract was also changed as suggested.
  • In “within 48 hours” group there were no samples collected less than 24 hours after therapy start, making the achieve of steady state very unlikely. We better explained this in methods.
  • We added a ref to teicoplanin.
  • added ref to vanco PK. Better specified that half-life is in adults. Protein binding is variable but does not exceed 50-55% also in pediatric patients.
  • Creatinine was measured by enzymatic IDMS validated method and if there were multiple dosages in the same day, the worst was considered. We added this information.
  • Our idea is to evaluate glomerular function with the version modified by Pottel et al (Pottel et al., "On the Relationship between Glomerular Filtration Rate and Serum Creatinine in Children."). This “Flanders metadata” is substantially a Schwartz formula where the constant K is modified in an age dependent way. As cited in Pottel's paper "Measuring and Estimating Glomerular Filtration Rate in Children." the formula thus modified is applicable to patients of a younger age than the modified Schwartz because the cohort of patients from which it was extracted consisted of healthy patients even less than one year of age. As the author suggests, however, this formula may show bias in patients older than 2 years so we decided to propose it for the assessment of eGFR only in infants, a population in which even the Schwartz modified formula does not show such a surprising performance. This concept was then better specified in methods.
  • In our institution, PICU and NICU both admit infants but for different problems: for example, cardiopathic infants are admitted to PICU while premature infants without other problems are managed in NICU. Since it was not possible to obtain PMA from our data source, we have, for simplicity, chosen to divide the case history according to PNA. We believe that, while simplistic and far from ideal, this approach showed a consistent difference in the two populations.
  • in neonatal population, a significant linear correlation was only found between age (in days) and eGFR (as age increases, GFR, an expression of renal maturation, increases). However, there is no correlation between eGFR and AUC (estimated with both formulas) in this population. These data have been better explained in the text.
  • The concept was better explained thanks to the paper suggested by reviewer on free fractions of vancomycin.

We hope that with the suggested changes our work can then be suitable for publication.

Best regards

Marcello Mariani

Round 2

Reviewer 3 Report

Comments and Suggestions for Authors

after re-reading ref 12 (the Pottel Flanders), i still miss the rationale to use this in newborns. To the best of my understanding, this formula has never been explored nor validated in newborns. Please proof me otherwise if I'm wrong, but the subsequent claimed hyperflitration is in my opinion incorrect, as likely rather maturational changes in clearance, not hyperclearance. 

If the claimed hyperclearance would be correct (>145 ml/kg/1.73m2), than vanco clearance should also be 'similar' hese adult values, and the authors do not come to this conclusion, and nor is there any evidence on this in the litature 

sorry for my persistence on this, I agree on the target attainment issues, but this is in my assessment not due to hyperfiltration, but rather relative underdosing (and early sampling, before steady state has been reached)

Author Response

We apologise to the reviewer for not immediately removing the Flanders metadata formula. After a better review of the literature and after discussions with other authors as well, we agree with the reviewer to use only Schwartz's formula. All references to Flanders metadata have therefore been removed and we have improved the discussion regarding the neonatal part: we too agree that in this population the lack of target attainment is due not only to variation in protein binding of vancomycin, but also probably to insufficient starting dosage of vancomycin. Probably the factor that (might) be less influential is the failure to reach steady state, as renal clearance in these patients is lower and therefore it should (theoretically) be reached more quickly.

Best regards

Marcello Mariani, MD